# New Monoclonal Antibodies Specific for Different Epitopes of the Spike Protein of SARS-CoV-2 and Its Major Variants: Additional Tools for a More Specific COVID-19 Diagnosis

**DOI:** 10.3390/biomedicines11020610

**Published:** 2023-02-18

**Authors:** Sabrina Mariotti, Maria Vincenza Chiantore, Raffaela Teloni, Angelo Iacobino, Antonio Capocefalo, Zuleika Michelini, Martina Borghi, Melissa Baggieri, Antonella Marchi, Paola Bucci, Silvia Gioacchini, Raffaele D’Amelio, Philip J. M. Brouwer, Silvia Sandini, Chiara Acchioni, Marco Sgarbanti, Antonio Di Virgilio, Felicia Grasso, Andrea Cara, Donatella Negri, Fabio Magurano, Paola Di Bonito, Roberto Nisini

**Affiliations:** 1Dipartimento di Malattie Infettive, Istituto Superiore di Sanità, 00161 Roma, Italy; 2Dipartimento Sicurezza Alimentare, Nutrizione e Sanità Pubblica Veterinaria, Istituto Superiore di Sanità, 00161 Roma, Italy; 3Centro Nazionale per la Salute Globale, Istituto Superiore di Sanità, 00161 Roma, Italy; 4Dipartimento di Medicina Clinica e Molecolare, Sapienza Università di Roma, 00161 Roma, Italy; 5Department of Medical Microbiology, Amsterdam UMC, University of Amsterdam, 1081 Amsterdam, The Netherlands; 6Centro per la Sperimentazione ed il Benessere Animale, Istituto Superiore di Sanità, 00161 Roma, Italy

**Keywords:** SARS-CoV-2, COVID-19, monoclonal antibody, spike protein, epitope mapping, diagnosis, rapid antigenic test, variants of concern

## Abstract

The emergence of the new pathogen SARS-CoV-2 determined a rapid need for monoclonal antibodies (mAbs) to detect the virus in biological fluids as a rapid tool to identify infected individuals to be treated or quarantined. The majority of commercially available antigenic tests for SARS-CoV-2 rely on the detection of N antigen in biologic fluid using anti-N antibodies, and their capacity to specifically identify subjects infected by SARS-CoV-2 is questionable due to several structural analogies among the N proteins of different coronaviruses. In order to produce new specific antibodies, BALB/c mice were immunized three times at 20-day intervals with a recombinant spike (S) protein. The procedure used was highly efficient, and 40 different specific mAbs were isolated, purified and characterized, with 13 ultimately being selected for their specificity and lack of cross reactivity with other human coronaviruses. The specific epitopes recognized by the selected mAbs were identified through a peptide library and/or by recombinant fragments of the S protein. In particular, the selected mAbs recognized different linear epitopes along the S1, excluding the receptor binding domain, and along the S2 subunits of the S protein of SARS-CoV-2 and its major variants of concern. We identified combinations of anti-S mAbs suitable for use in ELISA or rapid diagnostic tests, with the highest sensitivity and specificity coming from proof-of-concept tests using recombinant antigens, SARS-CoV-2 or biological fluids from infected individuals, that represent important additional tools for the diagnosis of COVID-19.

## 1. Introduction

The severe acute respiratory coronavirus 2 (SARS-CoV-2) has been identified as the pathogen responsible for coronavirus infectious disease 2019 (COVID-19) [1]. SARS-CoV-2 is an enveloped virus with a positive, capped and polyadenylated, single-stranded RNA genome of approximately 30 kb. SARS-CoV-2 belongs to the *Betacoronavirus* genus in the *Coronaviridae* family [2]. The genomic RNA has at least 10 open reading frames (ORF). ORF1a and ORF1b, produced by ribosomal frameshifting, code for two long polyproteins, pp1a and pp1ab, processed in 16 non-structural proteins (nsp1–nsp16), comprising viral enzymes such as the RNA-dependent RNA polymerase (RdRp) and two viral proteases (PL proteinase and 3CL). The non-structural proteins that rearrange, within the rough endoplasmic reticulum (ER) and Golgi compartments’ membranes, into double-membrane vesicles wherein viral replication and transcription occur (viral factory) [3]. The entire replication cycle takes place in the cytoplasm. One-third of the SARS-CoV-2 genome encodes four main structural proteins from subgenomic RNAs coded in the order: spike (S), envelope (E), membrane (M), and nucleocapsid (N) proteins. Several small accessory proteins (3, 6, 7a, 7b, 8, and 9b) are coded in this region, some with essential functions for the virus life cycle [4,5,6]. 

SARS-CoV-2 employs a receptor-binding motif within the S protein for binding the host angiotensin-converting enzyme 2 (ACE2) receptor for cell entry [7]. The S-ACE2 binding process is followed by a proteolytic cleavage of S by a plasma membrane-anchored serine protease 2 [7]. This activation results in conformational changes that allow fusion of the viral membrane with the host cell membrane and the RNA genome to enter the cytoplasm [8,9].

On 11 March 2020, the World Health Organization (WHO) officially declared COVID-19 to be a pandemic. The emergence of virus variants with increased infectivity or with potential antigenic escape ability (B.1.1.7, B.1.351, P.1, B.1.617, and B.1.1.529) possibly contributed to the rise of infections that nowadays count over 664 million confirmed cases with over 6.7 million deaths worldwide (Accessed on 24 January 2023: https://covid19.who.int). The pandemic had a devastating impact on the global economy and public health systems worldwide. The availability and large-scale administration of vaccines for COVID-19 represented a significant step forward in the fight against SARS-CoV-2 [10]. However, monoclonal antibodies (mAbs) able to recognize all major virus variants are still needed as diagnostic tools for antigenic rapid tests to be performed with saliva or nasal swabs for identifying infected individuals to be treated or quarantined [11,12]. Numerous assays are now commercially available, but the epidemiology changes and the rapid spread of SARS-CoV-2 variants worldwide has caused a renewed need for adjunctive reagents for diagnosis and immunotherapy [13].

The first generation of antigenic diagnostic tools greatly contributed to the control of the disease spread, and the vaccination has dramatically reduced the number of severe cases of COVID-19 and the associated deaths [14]. However, the number of infected people is still high worldwide, and an accurate diagnosis is still urgently needed. With a desirable reduction in cases expected in the near future and downsizing of the pandemic’s spread, more specific tests will be required in order to discriminate symptoms related to SARS-CoV-2 from those caused by other coronaviruses. The technical European Commission working group on COVID-19 diagnostic tests (https://health.ec.europa.eu/health-security-and-infectious-diseases/crisis-management/covid-19-diagnostic-tests_en#contact-information, accessed on 30 January 2023) listed the authorized available antigenic tests validated through prospective (Category A) or retrospective (Category B) clinical trials. The majority of these tests use antibodies against the N protein of the virus, and only in 1.48% of the tests were anti-S antibodies used to detect SARS-CoV-2 proteins in biofluids. SARS-CoV-2 rapid tests based on the detection of the N protein cross-reacted with SARS-CoV [15,16] and, particularly if using polyclonal anti-N antibodies elicited in rabbit, these tests have a high probability to recognize the N protein of other alpha or beta human common coronaviruses, which share up to 40% of their amino acid sequence with SARS-CoV-2 and show potential common linear as well as conformational epitopes [17]. In a previous work, we described the generation of mAbs specific for the receptor binding domain (RBD) of SARS-CoV-2 [18]. These mAbs recognized conformational epitopes and were extremely efficient in neutralizing the virus, but their possible use as diagnostic tools could be limited by the possible mutation in the RBD regions of new variants of concern (VOC) under immunological pressure [19,20]. With these premises, we thought that the availability of mAbs specific for constant regions of the S protein of SARS-CoV-2 could represent a significant step forward and contribute to improving the antigenic diagnosis of COVID-19. 

In this study, we immunized mice with a recombinant S protein in order to obtain mAbs able to bind conserved epitopes of S protein potentially useful for a more specific antigenic diagnosis of COVID-19.

## 2. Materials and Methods

### 2.1. Cloning, Expression and Purification of SARS-CoV-2 Proteins in E. coli

For the expression in *E. coli*, DNA sequences coding for S1 (14–682 aa) and S2 (698–1196 aa) were amplified by polymerase chain reaction (PCR) from codon-optimized synthetic DNA sequence (GenScript, Leiden, the Netherlands) encoding the SARS-CoV-2 S protein of Wuhan-Hu-1 isolate (NCBI reference sequence: NC_045512.2) with the opportune primers (Table 1). The 698–1196 S2 segment was chosen according to the S2 *E. coli* recombinant protein commercially available at that time (RayBiotech, Norcross, GA, USA; [21]). The obtained amplicons were cloned into the pGEM-T (Promega, Madison, WI, USA) intermediate vector and subcloned in pQE30 (Qiagen, Hilden, Germany) into BamHI /HindIII restriction sites. The proteins have an RGS(H)_6_ tag at the N-terminus and were purified by Ni-NTA affinity chromatography on either Cytiva HisTrap Excel columns or Ni-NTA-agarose (Qiagen) in batch. The proteins were purified by following a denaturing protocol. In order to optimize the yield of each fragment, the handbook QIAexpressionist was followed. The proteins were quantified in densitometry SimplyBlu-stained bands of sodium-dodecyl sulphate poly-acryl-amide gel electrophoresis (SDS-PAGE) through comparison with known concentrations of bovine serum albumin (BSA). The proteins were stored in elution buffer containing urea for their use in enzyme-linked immunosorbent assay (ELISA) and Western blot (WB). The denaturing buffer was removed by dialysis with an opportune buffer when necessary. The proteins were identified by WB using the monoclonal RGS·His antibody (Qiagen) and polyclonal anti-S SARS-CoV-2 antibodies described below.

### 2.2. Cloning, Expression and Purification of SARS-CoV-2 Proteins in HEK293T Cells

Three constructs of several domains of S protein were generated using pSecTag2Hygro A as a vector, which was designed for high-level expression and secretion in mammalian cells. Briefly, pSecTag2Hygro A contains the cytomegalovirus (CMV) promoter for high-level constitutive expression; an immunoglobulin kappa light chain (Igk) leader sequence specifying the secretion of heterologous proteins and a C-terminal polyhistidine (6xHis) tag for the rapid purification by Ni-NTA affinity chromatography. In order to enhance the expression of the S protein of SARS-CoV-2 in mammalian cells, a codon-optimized cDNA encoding the S protein was synthesized by GenScript Biotechnology. In order to achieve the expression of the S protein in a soluble form by mammalian cells, the full-length coding sequence for SARS-CoV-2 S protein encompassing residues 16–1182 and lacking the transmembrane domain was amplified with a primer pair including an HindIII restriction site in the 5′ end in frame with the Igk signal peptide and an XhoI restriction site in the 3′ end in frame with the His Tag of the plasmid backbone (pIgkS_16-1182_ΔTM). In order to achieve the expression of soluble forms of the main domains of the S, the coding sequence for S1 (16–682 residues) and S2 (685–1182 residues) domains were amplified and sub-cloned in the pSecTag2Hygro A plasmid using the same cloning strategy to generate the expression plasmids pIgKS1_16-682_ and pIgKS2_685-1182_, respectively. 

The expression of recombinant proteins was carried out in the HEK293T cell line using polyethylenimine (PEI) transfection reagent (Sigma, Kawasaki, Japan) as previously described [18]. Briefly, cells were seeded with 1 × 10^7^ cells in a 175 Flask and incubated overnight at 37 °C in 50 mL of Advanced DMEM containing 2% fetal bovine serum (FBS; LONZA). The culture medium was then replaced with 45 mL of Ham’s F-12 Nutrient Mixture (F-12, Gibco) supplemented with a transfection mix composed of 5 mL of Opti-MEM Reduced-Serum Medium (Gibco, Waltham, MA, USA) containing 87.5 µg of plasmids and 87.5 µL of PEI (0.5 mg/mL), and after 72 h of incubation, the supernatant was purified. Supernatant pH was adjusted to 8.0 with the equilibration buffer (0.2 M sodium phosphate, 300 mM NaCl, 20 mM imidazole, and 0.1% Tween20, pH 8.0), combined with 1 mL of 50% slurry Ni-NTA agarose (Qiagen) and gently agitated overnight at 4 °C. The supernatant was then loaded onto the column, and the proteins were eluted in 0.1 M sodium phosphate, 300 mM NaCl, 250 mM imidazole, and 0.05% Tween20, pH 8.0. Fractions containing the protein were pooled and dialyzed in PBS buffer. The quantification of proteins produced in the mammalian system was achieved by resolving on SDS-PAGE (see below) increasing volumes of purified recombinant (r) S1 or rS2, and using known concentrations of BSA as standard. The gels were stained with a fluorescent protein stain (Krypton Protein Stain, Thermo Scientific, Waltham, MA, USA) according to the manufacturer’s instructions and were analyzed using ChemiDoc MP Imaging System (Biorad, Hercules, CA, USA) and Image Lab Software (Image Lab 6.0.1). 

HEK293T cells (5 × 10^6^ cells) were seeded on 10 cm Petri dishes (Corning Incorporated) and transiently transfected with 10 µg of plasmids of S protein of HCoV-OC43, MERS-CoV, SARS-CoV, HCoV-NL63, and HCoV-229E using PEI transfection reagent in 10 mL of F-12. At 48 h post transfections, cells were collected and lysed for WB analysis. A schematic representation of the procedure is reported in Appendix A.

### 2.3. N-Deglycosylation of rS1 and rS2 Proteins

N-deglycosylation of HEK293T-produced rS1 and rS2 was performed using the peptide-N (4)-(N-acetyl-b-D-glucosaminyl) asparagine amidase F (PNGase F; Roche, Basel, Switzerland) according to manufacturer’s instructions. Briefly, S proteins were denatured for 10 min at 95 °C, followed by the addition of PNGase F (1 U/50 ng) and incubation at 37 °C for 30 and 60 min. Deglycosylated samples were diluted in SDS-loading buffer and incubated at 60 °C for 15 min, then they were analyzed by SDS-PAGE followed by WB using anti-Tetra His mAb (Qiagen).

### 2.4. Cells and SARS-CoV-2 Virus Propagation

Vero E6 (ATCC# CRL-1586) cells were maintained in Dulbecco’s modified Eagle’s medium (DMEM, Gibco) supplemented with 10% heat-inactivated FBS, 2 mM L-glutamine, 1 mM sodium pyruvate, and 1x non-essential amino acids (Gibco) without antibiotics or antimycotics. 

Viral isolates Wuhan (BetaCov/Italy/CDG1/2020|EPI ISL 412973|2020-02-20; GISAID accession ID: EPI_ISL_412973), Alpha (SARS-CoV2/hCoV-19/England/204820464/2020/#NR54000 (P4) 7-06-2021 (BEI Resources, Newport News, VA, USA), Beta (SARS-CoV2/hCoV-19/South Africa/KRISP-EC-K005321/2020/#NR54008 (P4) 7-06-2021 (BEI Resources), Delta (hCoV-19/Italy/ISS/B.1.617.2_2021), and Omicron (hCoV-19/Italy/ISS/BA.1_2021) VOC were propagated through the inoculation of 70% confluent Vero E6 cells in 75 cm^2^ cell culture flasks. Vero E6 cells were inoculated with SARS-CoV-2 aliquot stored at −80 °C at a multiplicity of infection of 0.01 in DMEM supplemented with 2% FBS. After an adsorption period of 1 h at 37 °C, the medium was replaced and the virus removed, and the cells were observed for cytopathic effects every 12 h. Stock SARS-CoV-2 virus and VOC were harvested at 72 h post infection, and supernatants were collected, clarified, aliquoted, and stored at −80 °C. Virus was quantified by plaque assay on Vero E6 cells.

### 2.5. SDS-PAGE and WB Analysis

Protein samples were lysed in sodium dodecyl sulfate (SDS)-loading buffer containing 50 mM Tris-HCl, pH 6.8, 5% beta-mercaptoethanol, 3% SDS, 50% glycerol, and 0.5% bromophenol blue. Samples were heated at 95 °C for 5 min and loaded onto 4–20% or 4–15% gradient mini-PROTEAN TGX precast gels (Biorad). In order to monitor protein purification, gels were stained with SimplyBlue^TM^ SafeStain (Novex, LifeTechnologies, Carlsbad, CA, USA). 

SDS-PAGE-gels were blotted onto Nitrocellulose (0.22 nm) or PVDF (0.45 nm) membranes either in Trans-Blot^®^ Cell in buffer containing 25 mM *Tris*, 192 mM *glycine*, pH 8.3, 20% *methanol* or in a Trans-Blot Turbo semidry system (Biorad) using a transfer kit and following the manufacturer’s instructions. Transfer efficiency was monitored by Ponceau S staining of the membranes or by staining of the gel after blotting. 

Membranes were incubated in blocking solution containing 3–5% skim milk (Millipore, Burlington, MA, USA) in Tris-buffered saline (25mM Tris-HCl pH7.4, 0.5 mM NaCl, TBS) for 2 h. In order to detect the S proteins, the following commercial antibodies, in TBS containing 0.05% Tween (TBST), were used: (1) polyclonal SARS-CoV-2 S antibody, Rabbit Pab, Sino Biological; (2) polyclonal SARS S Protein Antibody NB100-56578T, Biotechne; (3) monoclonal Tetra HIS Antibody (Qiagen). Filter-bound immunoglobulins were detected by Horseradish Peroxidase-conjugated goat anti-mouse or anti-rabbit IgG (H + L) (ABCAM) by either Crescendo Western HRP chemiluminescent substrate (Millipore) or TMB colorimetric substrates (Vector, Olean, NY, USA).

Vero E6 cells’ supernatant containing 1 × 10^6^ plaque-forming units (PFU)/mL of SARS-CoV-2 Wuhan, Alpha, Beta, Delta, or Omicron VOC was inactivated for 1 h at 56 °C and then lysed with opportune SDS-loading buffer containing beta-mercapto-ethanol. Lysates were loaded on preparative SDS gels, which were blotted onto PVDF, and each filter was cut into 10strips. Each strip was incubated with a specific S1 or S2 mAb. 

Cell lysates of HEK293T cells transiently transfected with plasmids (SinoBiological, Beijing, China) expressing the S protein of SARS-CoV-2 (pCMV3-SARS-CoV2-Spike-His), SARS-CoV (pCMV3-SARS-CoV-Spike-Flag), HCoV-OC43 (pCMV3-HCoV-OC43-Spike-Flag), MERS-CoV (pCMV3-MERS-CoV-Spike-Flag), HCoV-NL63 (pCMV3-HCoV-NL63-Spike-Flag), and HCoV-229E (pCMV3-HCoV-229E-Spike-Flag) were subjected to SDS-PAGE followed by WB with anti-S1 or anti-S2 mAbs. 

Nasal swabs of individuals negative or positive for SARS-CoV-2 infection (whose diagnosis had been confirmed by rRT-PCR) were lysed with RIPA buffer, and, after the addition of SDS loading buffer containing beta-mercapto-ethanol, were subjected to SDS-PAGE followed by WB with the specific mAb.

### 2.6. Mice Immunization

Six-week-old female pathogen-free BALB/c mice were obtained from Charles River Laboratories (Calco, LC, Italy) and housed in the Istituto Superiore di Sanità. All animal protocols and procedures were performed in accordance with European Union guidelines and Italian legislation (DL26/2014) and have been approved by the Italian Ministry of Health and reviewed by the Service for Animal Welfare at ISS (Protocol Number: # 670/2020-PR of 21 July 2020). Mice were subcutaneously immunized with 100 µL/mouse solution containing 5 µg of HEK293T rS protein ectodomain (aa 1–1138 of SARS-CoV-2 S) on both sides of the lower anterior abdomen on days 0, 14, and 28, as previously described [18]. The rS protein ectodomain, which had a mutated furin cleavage site (RRAR to GGGG) and stabilizing double proline (2P) substitutions [22], was produced at the Amsterdam University Medical Center.

For priming, the antigen was mixed with an equal volume of emulsified complete Freund’s adjuvant (Millipore-Sigma) immediately prior to administration. For boosts, the recombinant protein was emulsified with an equivalent volume of incomplete Freund’s adjuvant (Millipore-Sigma). Serum samples were collected on days 0 (pre-immunization), at day 14 (before the protein immunization), and at day 42 (2 weeks after the third immunization) by retro-orbital blood collection, and they were stored at −20 °C until analysis. The collected plasma samples were tested by in-house ELISA for the determination of anti-S specific antibody titles in order to identify the high responder mouse. The spleen from the mouse showing the highest titer of specific antibody was selected for the generation of mAbs.

### 2.7. Isolation and Purification of Specific mAb-Producing Clones

The spleens of selected mice were used to isolate S-specific mAbs as previously described [18]. Briefly, spleen cells were incubated with 5 mL lysis buffer on ice for 5 min, and after washing, were mixed with the mouse myeloma cell line SP2 in a 2:1 = splenocytes:myeloma cells ratio. They were then centrifuged, and the cell pellet was slowly resuspended with 1 mL of polyethilenglycol (PEG, Sigma-Aldrich, St. Louis, MI, USA). Afterwards, 7 mL of complete RPMI medium supplemented with 25 mM of Hepes was slowly added, and the centrifuged pellet was resuspended in complete RPMI medium containing selective hypoxanthine, aminopterin, and thymidine (HAT, Sigma-Aldrich) and cultured in a flat-bottomed 96-well plate for 14 days at 37 °C with 5% CO_2_. The fused growing cell lines were selected by microscope examination, transferred to new 96-well plates and expanded in complete RPMI medium containing selective HAT. Hybridomas were screened for antigen specificity by ELISA using HEK293T S protein-coated plates. The selected polyclonal Ab-producing hybridomas were single-cell-cloned by limiting dilution in the presence of 5 × 10^4^ cells/well feeder splenocytes in a flat-bottom 96-well plate in complete RPMI medium containing selective HAT for 12 days. The growing clones were tested for antigen specificity by ELISA. The clones producing S-specific mAbs were expanded in static T75 flasks in the presence of 50 mL of DCCM2 (Biological Industries, Cromwell, CT, USA) supplemented with kanamycin, and the mAbs were purified and concentrated using chromatography cartridges’ protein G columns (Thermo Fisher). The concentration of purified mAbs was evaluated by a NanoPhotometer (Implen, Münich, Germany) spectrophotometer at 280 nm. 

### 2.8. Nasal Swabs

Nasal swabs were from healthy or COVID-19-convalescent volunteers who gave their informed consent to participate in the collaborative study between Istituto Superiore di Sanità and the Italian Air Force entitled: “Valutazione della performance analitica di un test antigenico per il rilevamento di SARS-CoV-2, confronto con un test di screening molecolare” [13]. According to the epidemiology of SARS-CoV-2 variants in Italy at the time of sampling (April 2022), there is a high probability that the patients could be infected with the VOC Omicron, even if viral sequencing of the samples was not performed. 

### 2.9. ELISA

The antigen specificity of the immunized mouse sera and the hybridoma supernatants was analyzed by ELISA, as previously described [18]. Briefly, 0.5 µg/mL of SARS-CoV-2 recombinant S protein, S1 or S2 domains, or 10 µg/mL of peptides were coated overnight at 4 °C (50 µL/well). After washings, the plates were blocked with PBS+2% BSA, and then, 50 µL of serum or supernatant samples were incubated for 3 h at 37 °C. Secondary goat anti-mouse total Ig or IgG or anti-IgG subclass alkaline phosphatase (PA)-conjugated (Southern Biotech, Birmingham, AL, USA) or mouse anti-human IgG PA-conjugated (Invitrogen) were incubated for 1 h at 37 °C. Plates were developed by adding 100 µL/well of substrate4-Nitrophenyl phosphate disodium salt hexahydrate and stopped with 50 µL of 3N NaOH. Absorbance (405 nm) was measured, and the results were considered positive if the optical density (OD) was three times greater than the negative control. 

For the sandwich ELISA, plates were coated overnight with 2 µg/mL of primary mAb (S200, S178, S71, S79) and then incubated with scalar doses of S recombinant protein (from 50 to 0.39 ng) for 1 h at 37 °C. A secondary (II) mAb, of a subclass different from the subclass of the primary mAb, was added (1 h at 37 °C), and then the sandwich was revealed with an antibody that recognizes the subclass of the II mAb. The most efficient pair (S71 and S79) was tested in a sandwich ELISA with a scalar quantity of heat-inactivated Wuhan SARS-CoV-2 (from 25000 to 195 PFU/well).

### 2.10. Dot Blot Assay

Dot blot assay was performed by coating the PVDF membrane with the supernatant of Vero E6 cells infected with SARS-CoV-2 or lysed nasal swabs from COVID-19-positive patients (whose diagnosis had been confirmed by rRT-PCR) and from control healthy subjects. Detection was carried out by anti-S1 mAbs.

### 2.11. Lateral Flow Assay (LFA)

LFA was performed using strips of a commercially available nitrocellulose membrane (ab274103) containing a ‘test line’ (T line) of streptavidin able to bind the biotin-conjugated capture antibody which further binds the samples in combination with the detection antibody. The strips also contain a ‘control-line’ (C line) of immobilized anti-mouse antibody, which shows that the test is valid. The capture and the detection mAbs were diluted in Tris-Glycine SDS Running Buffer 1X with BSA 0.1%, along with the samples, the supernatant of VeroE6 cells infected by SARS-CoV-2, and the lysed nasal swabs from individuals negative or positive for COVID-19. For a single strip, a mix of mAbs and sample was prepared, incubated for 5 min, and loaded into a 96-well plate. A strip was inserted into each well, and the mixture was run for 20 min. Strips were then incubated with isotype-specific alkaline phosphatase (AP)-conjugated anti-mouse Ab. The T and the C lines were detected by AP substrate (Roche), following the manufacturer’s instructions. 

### 2.12. Plaque Reduction Neutralization Test (PRNT)

PRNT was conducted as previously described [23]. A volume (300 µL) of each purified mAb under serial dilution starting from 10 μg/mL was incubated with 80 PFU of SARS-CoV-2 at the final volume of 600 µL at 4 °C overnight. The mixtures were added in triplicates to confluent monolayers of Vero E6 cells, grown in 12-well plates, and incubated at 37 °C in a humidified, 5% CO2 atmosphere for 60 min. Then, 4 mL/well of a medium containing 2% Gum Tragacanth (Sigma Aldrich) + MEM 2.5% FCS were added. Plates were left at 37 °C with 5% CO_2_. After 3 days, the overlay was removed, and the cell monolayers were washed with PBS in order to completely remove the overlay medium. Cells were stained with a crystal violet 1.5% alcoholic solution. The presence of SARS-CoV-2 virus-infected cells was indicated by the formation of plaques. The inhibitory concentration (IC)50 was determined as the highest dilution of serum resulting in a 50% (PRNT50) reduction of plaques as compared to the virus control.

### 2.13. Epitope Mapping

A selection of synthetic peptides of 40 aa in length (with 20 aa overlaps between sequential peptides) was obtained by Bio-Fab Research (Rome, Italy) and is listed in Table 2. After dissolving these peptides with suitable solvent (depending on the given peptide’s solubility), they were used to coat plates at 10 µg/mL in an in-house ELISA performed as described above.

### 2.14. Statistical Analysis

For the neutralization and ELISA experiments, technical duplicates were performed. All of the statistical analyses were performed using the GraphPad Prism software v9 (GraphPad Software, San Diego, CA, USA). The half maximal effective concentration (EC_50_) was calculated through the non-linear regression analysis of the log10 of serum dilution plotted versus the absorbance at 405 nm. 

## 3. Results

### 3.1. Production of SARS-CoV-2 Recombinant Proteins

SARS-CoV-2 recombinant proteins were produced in both eukaryotic (HEK293T) and prokaryotic (*E. coli*) expression systems. Three constructs of the S protein were generated using a mammalian cell codon-optimized sequence encoding the ectodomain of SARS-CoV-2 spike protein Wuhan-Hu-1 isolate (NCBI reference sequence: NC_045512.2) as a template for high-level expression and secretion in mammalian cells: the full length of the coding sequence for the SARS-CoV-2 S protein encompassing residues 16–1182 and lacking the transmembrane domain (pIgkS_16-1182_ΔTM), the coding sequence for S1 (16–682 residues, pIgKS1_16-682_) and for S2 (685–1182 residues, pIgKS2_685-1182_) (Figure 1A). The expression of the soluble forms of S1, S2, and SΔTM was evaluated in the supernatant of HEK293T cells by WB (Figure 1B). In the SDS-PAGE analysis, our recombinant proteins migrated as products of more than their theoretical mass deduced from the amino acid sequence (79.0 kDa, 58.7 kDa, 133.5 kDa for rS1, rS2, and S16–1182ΔTM, respectively) due to the existence of glycosylation. Moreover, the secreted S_16-1182_ΔTM recombinant protein was cleaved by Furin-like protease. The recombinant proteins produced in *E. coli* (Figure 1C) showed the expected molecular mass of 75 kDa for S1 (14–682 residues) and 60 kDa for S2 (698–1196 residues) (Figure 1D). 

The recombinant proteins S1 and S2 produced in HEK293T cells and used for the screening of mAbs were checked by Krypton-stained SDS-PAGE that showed the absence of contaminants (Figure 1E) and allowed for their quantification. 

The SARS-CoV-2 S glycoprotein contains 22 N-linked glycosylation sequons per protomer [24]. The presence of N-linked glycosylation in rS1 and rS2 was confirmed by analyzing the molecular weight of the recombinant proteins before and after enzymatic treatment with the glycosidase PNGase F. The time-dependent decrease of the rS1 and rS2 molecular mass in correlation with PNGase F exposure indicated that the proteins were glycosylated (Figure 1F). 

### 3.2. Production of SARS-CoV-2 Spike-Specific mAbs

The procedure used to produce hybridomas was highly efficient, and among 40 different specific mAbs isolated, we purified and characterized the 13 mAbs that were able to bind the recombinant S protein produced in HEK293T with the highest efficiency in ELISA. Of the 13 selected mAbs, 9 recognize the S1 domain, and 4 were reactive with the S2 portion, as assessed by ELISA (Figure 2A) and confirmed by WB (Figure 2B). In order to evaluate the binding affinity of the selected mAbs, we performed an ELISA with scalar doses of rS1 or rS2 proteins (Figure 2C): the EC_50_ required for all mAbs to bind recombinant proteins was below 100 ng/mL for all of the S2-specific clones and for half of the clones reactive to the S1 domain (Table 3), indicating a high binding affinity. The anti-S mAbs purified on protein G columns were further characterized for Ig class and subclasses. All of the selected mAbs were IgG, and specifically, 10 clones were IgG1, 1 clone was IgG2a, and 2 clones were IgG2b (see Table 3). 

All of the S1-specific mAbs and one mAbs specific for the S2 domain (S3) indistinctly recognize proteins made with both production systems, while 3 of the 4 clones reactive to S2 (S136, S222, and S223) recognize only the protein produced in eukaryotic cells (Figure 3A). This discrepancy could be dependent upon post-translational modifications of the protein produced in HEK293T cells, or to the different lengths of the S2 proteins produced in *E. coli*. Glycosylation is a post-translational modification present in proteins produced in mammalian cells but absent in proteins produced in *E. coli*. In order to evaluate the involvement of N-glycosylation in the antigen recognition of S136, S222, and S223 mAbs, WB assay was performed using eukaryotic rS2 before and after the enzymatic treatment with glycosidase PNGase F. All three of the mAbs were able to bind both the glycosylated and the N-deglycosylated protein (Figure 3B). We hypothesized that these three mAbs bind S2 in O-glycosylated epitopes or in the region defined by aa 686–698, that is not present in S2 produced in *E.coli,* and therefore, we synthesized peptides encompassing this amino acid sequence (see below and Table 2 and Table 3) in order to settle this issue. 

### 3.3. Characterization of Epitope Recognized by Isolated S-Specific mAbs

An ELISA analysis was performed using 24 synthetic peptides (listed in Table 2) of 40 aa in length (with 20 aa overlaps between the sequential peptides), covering the S1 domain and the N-terminal portion of the S2 domain in order to achieve a more specific characterization of the epitope recognized by our anti-S mAbs. Table 3 and Figure 4A show the peptides recognized by the selected mAbs with an accuracy of 14 aa, 20 aa, or 30 aa, depending on positive results being obtained from single or overlapping peptides. mAb S200 recognized an epitope encompassing 20 aa in the N-terminal region of the S1 domain (aa 21–40). Six mAbs localized nearby the RBD: S37 binds the aa in position 281–300, while the two mAbs S75 and S106 recognize the same aa sequence at position 301–320. Three other mAbs (S71, S79, and S178) localize in the region downstream the RBD: the sequence of amino acids at position 550–580. However, it was possible to use S71 (IgG2b) and S79 (IgG2a) as an mAb couple in a sandwich ELISA and LFA (see below). This result suggests that they do not interfere with each other for antigen binding, indicating that they recognize different epitopes within the same aa sequence (550–580). The pair of mAbs S79 (IgG2a) and S178 (IgG2b) did not interfere in a sandwich ELISA, indicating that these two mAbs recognize different amino acids within the 550–580 sequence. On the other hand, we did not test S71 with S178, since they belong to the same IgG subclass and could not be used in a sandwich ELISA, so we cannot exclude that S71 and S178 bind to the same epitope. The final two mAbs that bind the S1 domain, S12, and S157, localize immediately upstream the S1/S2 cleavage site, in the sequence of aa at position 661–682. The 3 mAbs binding the S2 domain (S136, S222, and S223) recognize the sequence of aa 686–700, which is not present in the S2 produced in *E. coli*. Finally, the mAb S3, which binds to the recombinant S2 proteins produced in both HEK293T cells and *E. coli*, did not recognize any of the used peptides. Therefore, we can deduce that the epitope of S3 mAb encompasses the region defined by aa 721–1182, but we cannot indicate a more precise mapping.

We then used different pairs of mAbs, chosen on the basis of their different IgG subclass, to set up an ELISA able to detect the recombinant SARS-CoV-2 S protein in solutions as a first step for the development of an antigenic kit for the detection of SARS-CoV-2 S antigen in biofluids. We performed a sandwich ELISA with a panel of pairs of all the selected mAbs and scalar doses of the recombinant S protein. Figure 4B clearly shows that several pairs of mAbs were highly effective in detecting the presence of a very small quantity of recombinant protein.

### 3.4. Isolated S-Specific mAbs Specifically Bind SARS-CoV-2 but no Other Human Coronaviruses

The ability of the generated mAbs to recognize the proteins present in SARS-CoV-2 virus was further evaluated by performing WB assay with heat-inactivated supernatant of SARS-CoV-2-infected Vero E6 cells containing 1 × 10^6^ PFU/mL. Figure 5A shows that all of the generated mAbs specific for the S1 and S2 domains were able to recognize SARS-CoV-2 (Figure 5A).

The specificity of the generated mAbs to recognize the S protein of SARS-CoV-2 was confirmed by analyzing their reactivity with the S proteins of other human coronaviruses (HCoV). Cell lysates of HEK293T cells transiently transfected with plasmids expressing the S protein of HCoV-OC43, MERS-CoV, HCoV-NL63, HCoV-229E, and HCoV-HKU1 were subjected to SDS-PAGE, followed by WB using anti-S1 or anti-S2 mAbs. The generated mAbs did not react with any of the S protein of the other HCoV analyzed (Appendix A). 

Next, HEK293T cells were also transfected with a plasmid expressing SARS-CoV S protein. Cell lysate was subjected to SDS-PAGE, and WB was performed with each anti-S1 and anti-S2 mAb. The results show that three out of all of the generated mAbs recognize the S protein of SARS-CoV (S12, S157, and S3) (Figure 5B). None of the tested mAbs showed SARS-CoV-2 neutralization activity at the PRNT assay (data not shown).

Finally, using the most efficient pair of mAbs (S71-S79), we explored the capacity of a sandwich ELISA to detect SARS-CoV-2 using scalar dilutions of heat-inactivated virus (down to 139 PFU). Figure 5C shows that the S71-S79 mAbs pair is able to detect very small quantities of virus, down to 781 PFU.

### 3.5. Isolated S-Specific mAbs Recognize the Major SARS-CoV-2 Variants of Concern

With the worldwide progression of the COVID-19 pandemic, several new SARS-CoV-2 variants containing mutations in their S protein have been isolated, showing increased infectivity and the ability to cause disease in susceptible individuals. Clinical studies were designed to define whether immunization with vaccines based on the original SARS-CoV-2 Wuhan-Hu-1 S sequence may be sufficiently protective against these VOCs. Moreover, mAbs developed for diagnostic or therapeutic purposes may not be useful for individuals infected with these variants and/or affected by COVID-19. 

We analyzed the ability of our isolated mAbs to bind the S protein present in SARS-CoV-2 VOC by performing a WB assay with supernatant of Vero E6 containing 1 × 10^6^ PFU/mL of SARS-CoV-2 Alpha, Beta, and Delta VOC in comparison with the Wuhan isolate. The virus lysates were subjected to SDS-PAGE and WB with a specific mAb (Figure 6A). All of the anti-S1 and anti-S2 mAbs were able to bind the tested VOC. Each filter was next incubated with rabbit polyclonal antibodies against SARS-CoV N and M proteins [25] in order to ascertain the equal loading of virus lysates (Figure 6B).

We also tested the binding ability of the mAbs in supernatant of SARS-CoV-2-infected Vero E6 cells containing 1 × 10^6^ PFU/mL of Omicron VOC analyzed by WB. Figure 6C shows that all of the tested mAbs can recognize this VOC. 

The binding affinity of the mAbs was also analyzed in nasal swabs of individuals negative or positive for SARS-CoV-2 infection (whose diagnosis has been confirmed by rRT-PCR). The nasal swabs were lysed and resolved on the SDS-PAGE followed by WB with the specific mAb. The results show that the generated anti-S mAbs can recognize the presence of the SARS-CoV-2 in the nasal swab, as shown in Figure 6D using the anti-S1 S71 and S79 mAbs.

### 3.6. The Isolated mAbs Are Useful Tools for an Antigenic Diagnostic Assay: A Preliminary Proof-of-Concept

As a preliminary proof-of-concept for the possibility of using the generated mAbs as tools for the antigenic diagnosis of COVID-19 in rapid tests, we performed dot blot and lateral flow assay. SARS-CoV-2 from supernatant of infected Vero E6 cells or lysed nasal swabs from COVID-19-positive patients (whose diagnosis was confirmed by rRT-PCR) and from control healthy subjects were directly coated on membranes and then incubated with mAbs revealed by specific anti-mouse antibodies as described in M&M. All of the mAbs were able to detect the presence of SARS-CoV-2 in both in vitro-produced viral sample, even when highly diluted, and in the biological sample, indicating a good sensitivity (Figure 7A). The two most efficient pairs of mAbs (S71–S79 and S71–S12) were analyzed in LFA. In Figure 7B,C is shown a representative LFA performed with the two pairs of anti-S1 mAbs in order to detect the viral antigenic protein in both the supernatant of Vero E6 cells infected with SARS-CoV-2 and in the nasal swab from a COVID-19-positive patient.

## 4. Discussion

The rapid and precise diagnosis of infectious diseases is fundamental to initiating specific treatment for infected individuals and to control the spread of the disease with public health measures, including confinement and contact tracing. This general rule is particularly important for infections causing pandemics such as the SARS-CoV-2 causing COVID-19 [26]. Rapid antigenic tests followed by RT PCR confirmation significantly contributed to the control of COVID-19, particularly in the period of greater viral circulation [27]. These tests underwent progressive development, and the last generation was generally endowed with good performance in terms of sensitivity, specificity, and positive/negative predictive values [28,29]. The COVID-19 pandemics reached a peak of new cases in the period from January to March 2022 (https://covid19.who.int/, accessed on 24 January 2023) followed by a slow but consistent reduction in the subsequent months. With the reduction in the prevalence of COVID-19, a result of the decreased number of susceptible individuals caused by a high vaccination rate and the increased number of naturally immunized individuals, the performance of the current methods along with the positive/negative predictive values, could decline [30]. Moreover, in colder seasons, outbreaks of community-circulating HCoV are expected, which would require differential diagnosis with COVID-19. HCoV 229E, OC43, NL63, and HKU1 are considered to be relatively benign respiratory pathogens in humans leading to upper respiratory tract diseases [31]. HCoVs are seasonal pathogens that are widespread on all continents. Their frequency and clinical course vary significantly according to age, region, and genetic background [32,33,34]. 

SARS-CoV-2 shares up to 40% of the amino acid sequence of its N protein with other HCoVs, and it shows potential common linear as well as conformational epitopes [16]. The majority of the commercially available antigenic tests were developed using N protein as the main target. The HSC Technical Working Group (https://health.ec.europa.eu/health-security-and-infectious-diseases/crisis-management/covid-19-diagnostic-tests_en#contact-information, accessed on 30 January 2023) defined the COVID-19 antigen tests evaluated by prospective clinical field studies as “Category A” and those evaluated by retrospective in vitro studies as “Category B”. Up to July 2022, all of the 54 Category A rapid tests and 4 laboratory-based tests detect N antigens, and 145 out of 148 rapid tests and 1 laboratory-based test among Category B detect N antigens, while only 3 detect S protein.

Since antigenic tests based on the recognition of SARS-CoV-2 N protein might be at increasing risk of providing false-positive and low predictive values in the decreasing phase of the COVID-19 pandemic, we developed a series of anti-S protein mAbs and performed an in-depth characterization in order to evaluate their suitability as new reagents for the development of additional diagnostic tools.

We chose S protein as the target, since its sequence is more specific for SARS-CoV-2 than N protein. Although S protein is present in lower amounts on the virus and in the biological fluids of infected individuals, it is anyhow measurable, as suggested by previous studies by mass spectrometry [35], and its detection would indicate more specifically the diagnosis of COVID-19.

We produced the rS proteins in both the eukaryotic and prokaryotic systems, the latter by using denaturing conditions, to select those mAbs able to bind the recombinant proteins in ELISA in the putative natural conformation and glycosylation as well as in the linear form. The binding of mAbs to proteins produced in eukaryotic HEK293T cells guarantees that mAbs may also recognize conformational epitopes within the tertiary structure stabilized by disulfide bonds and glycosylation. On the other hand, the capacity to bind to the proteins produced in *E. coli* using denaturing conditions is indicative of the capacity of mAbs to bind to the linear sequence of the same antigen [36]. This is particularly interesting, since the use of the selected mAbs would detect both the virus and its released protein irrespective of the possible denaturation caused by human mucosal enzymes or by the procedures for the dilution and storage of samples taken from patients [37]. The binding of the selected mAbs was not prevented by N-glycosylations, which have been indicated as a possible mechanism of SARS-CoV-2 to immune-evade Ab recognition [38,39,40], since our mAbs recognized the fully glycosylated proteins produced in HEK293T cells and the in vitro de-N-glycosylation of the protein did not increase the mAbs binding.

We investigated the fine specificity of our mAbs, taking advantage of a peptide library spanning the sequence of the S1 and S2 according to a pre-screening based on protein S1 and S2 domain recognition. Using the chosen set of 40aa peptides (with 20aa overlapping), we were able to identify the epitope recognized by 12 out of the 13 selected mAbs. The peptide containing the epitope bound by the mAb S3 was not included in the designed set of peptides, and the binding site of mAb S3 could be approximatively indicated by inference due to its binding to the protein fragment S2 and not to the used peptides. The epitope mapping permitted one to foresee that our mAbs would recognize not only the S protein of the original Wuhan SARS-CoV-2 strain, but also its major variants, whose sequencing indicated the conservation of the epitopes recognized by our mAbs. In fact, all of the tested mAbs were able to bind to most of the VOC circulating and available for testing up to September 2022, as indicated by WB analysis. According to the sequence analysis, our mAbs may also be capable of recognizing the newest variants, including XBB1.5, which presents mutations not affecting the binding site of the mAbs. A possible exception is represented by mAb S200, which binds Omicron BA.1 but may not be able to recognize the newest variants that show some amino acid variations in their epitopes, such as the del24–26 and A27S mutations in BA.1 e XBB1.5.

Interestingly, none of the selected mAbs recognize common community HCoV, and three of them bind to the S protein of SARS-CoV. For COVID-19 screening purposes, we therefore selected mAbs S71 and S79, since they specifically recognize SARS-CoV-2 or its recombinant S protein with the highest efficiency and do not recognize HCoV, SARS-CoV, or MERS-CoV. These mAbs were indeed capable of recognizing the S protein of all of the tested SARS-CoV-2 variants, and they are presumably able to detect the newest Omicron variants, including XBB1.5. Should new variants with mutations to the epitopes recognized by these mAbs emerge, the knowledge of the sequence recognized by the other mAbs will present the chance to select and test some of the other mAbs for the rapid set-up of an updated screening test.

None of the tested mAbs showed SARS-CoV-2 neutralization capacity as assessed by PRNT. The observation that the binding to S protein by our mAbs does not interfere with virus entry into target cells suggests that the epitopes recognized by our mAbs are structural but not strategic for the viral life cycles. Therefore, it is reasonable that there is less evolutionary and immunologic pressure for mutations in these sequences [19,20], rendering our mAbs important for preparing diagnostic tools based on antigen detection that could be less affected by viral mutations.

We have tested the potential of these mAbs in dot blot and LFA as proof-of-concept of their suitability as diagnostic tools. However, the major limitation of this work is the lack of demonstration of the diagnostic potential of our mAbs in a controlled clinical trial. An LFA suitable for this trial is being developed, and the diagnostic capacity will be proven in comparison to commercially available tests based on N antigen detection.

## 5. Conclusions

This manuscript describes the functional characteristics of the set of mAbs that we have generated by immunizing mice with a recombinant S protein of SARS-CoV-2. 

We have analyzed in depth the characteristics of our mAbs, which may be proposed to develop new antigenic tests in addition to those based on the recognition of the SARS-CoV-2 N protein, which might be at increasing risk of providing false positive and low predictive values. We demonstrated the fine specificity for the S antigen of SARS-CoV-2 and its major variants and the lack of cross-reactivity of the described anti-S protein mAbs with other common community HCoVs. The absence of virus neutralizing capacity suggests that the epitopes recognized by our mAbs have less probability of undergoing mutations due to immunological pressure. Together, the data indicate that the described mAbs may be suitable for antigen detection in rapid or laboratory tests for the diagnosis of infection by SARS-CoV-2 and its actual and potential future variants. 

## Figures and Tables

**Figure 1 biomedicines-11-00610-f001:**
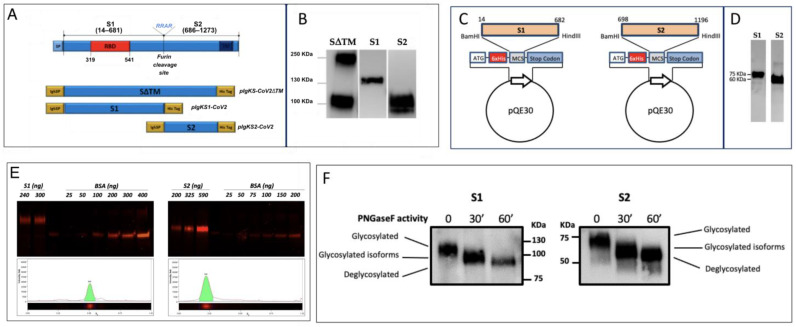
**SARS-CoV-2 S recombinant proteins.** (**A**) In the top diagram, the full-length SARS-CoV-2 S protein is depicted. SP, signal peptide; S1, subunit 1; S2, subunit 2; RBD, receptor-binding domain; TM, transmembrane domain. On the bottom, the diagram of the three different spike regions is shown, expressed by the generated plasmids: pIgkS16–1182ΔTM; pIgKS116–682; and pIgKS2685–1182. IgK SP: immunoglobulin kappa light chain signal peptide; His Tag: polyhistidine tag (6xHis). (**B**) WB analysis for the detection, using an anti-His tag, of the S ectodomain soluble form (SΔTM) and the recombinant S1 and S2 domains in the supernatants of transfected HEK293T cells. Strips of three different WBs are shown. (**C**) Schematic representation of S1 and S2 constructs in the *E. coli* expression vector. (**D**) The identities of proteins were confirmed by WB using the polyclonal anti-S SARS-CoV-2. (**E**) Upper: increasing quantities of rS1 or rS2 and known concentrations of BSA were resolved on SDS-PAGE, and quantification was achieved using Krypton fluorescent protein stain. Bottom: the electrophoretic profiles of recombinant proteins are reported. Lane 2 from both gels was selected as the representative of rS1 or rS2: the presence of a unique peak for both proteins indicates the absence of contaminants. (**F**) Purified rS1 and rS2 were denatured and digested with PNGase F for 30 and 60 min. The positions of glycosylated and deglycosylated proteins are indicated. A representative experiment of three is shown.

**Figure 2 biomedicines-11-00610-f002:**
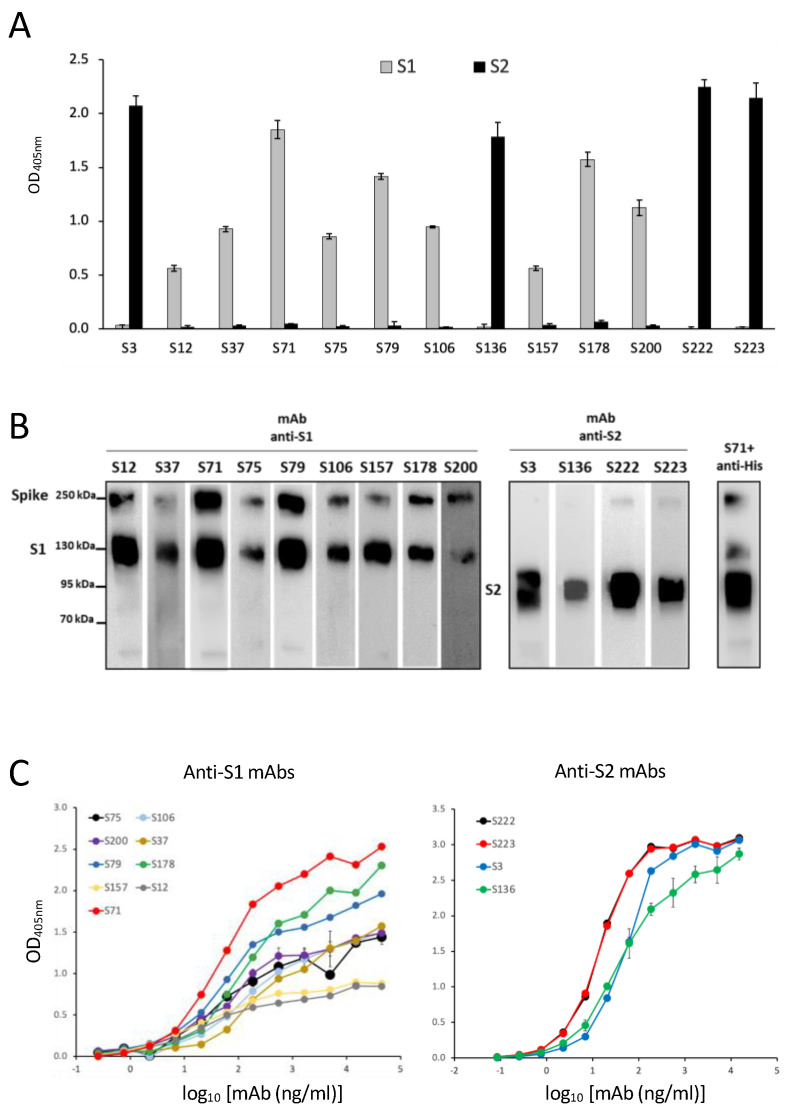
**Isolated mAbs specific for SARS-CoV-2 Spike.** (**A**) The binding of selected mAbs to the recombinant S1 or S2 domain produced in the mammalian expression system was evaluated by ELISA. Error bars indicate the standard deviations of technical duplicates from a representative experiment repeated three times. (**B**) WB analysis of anti-S1 and anti-S2 mAbs in SDS-PAGE supernatants of HEK293T cells transfected with pIgkS16–1182ΔTM. Anti-His antibody combined with anti-S1 S71 mAb was used as a positive control. (**C**) ELISA-binding affinity of the selected purified mAbs to the S1 (left panel) or to the S2 (right panel) domains produced in a mammalian expression system. Error bars indicate the standard deviations of technical duplicates from a representative experiment repeated twice.

**Figure 3 biomedicines-11-00610-f003:**
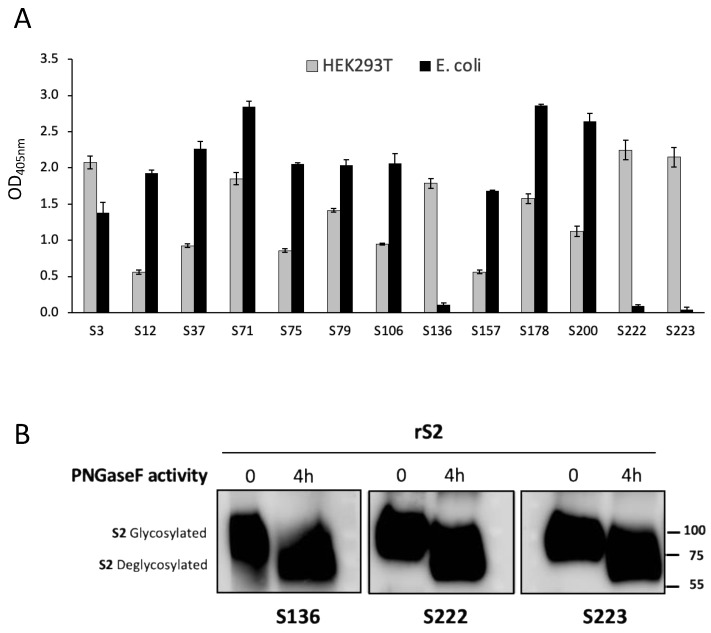
**mAbs specific biding to rS1 or rS2 produced in HEK293T or in *E. coli*.** (**A**) The binding of selected mAbs to the specific recombinant domain produced in mammalian (HEK293T) or prokaryotic (*E. coli*) expression systems evaluated by ELISA. Error bars indicate the standard deviations of technical duplicates from a representative experiment repeated three times. (**B**) WB analysis of anti-S2 mAbs S136, S222, and S223. In SDS-PAGE rS2 before and after glycosidase PNGase F treatment for 4 h. The positions of glycosylated and deglycosylated protein are indicated. A representative experiment of two is shown.

**Figure 4 biomedicines-11-00610-f004:**
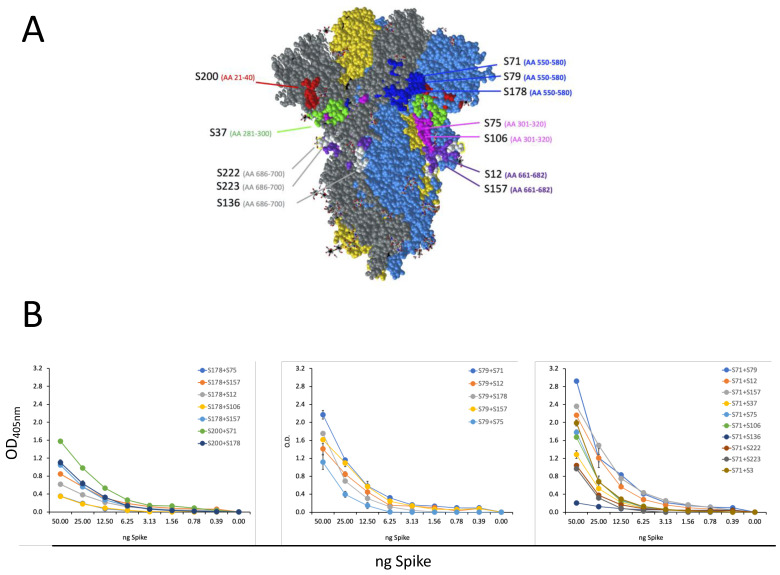
**Characterization of epitope recognized by isolated S-specific mAbs.** (**A**) Epitope mapping on the 3D structure of trimeric spike protein (retrieved from PDB 6VXX, [9]). Each monomer of the trimeric spike is represented in grey, blue, or yellow, respectively. Different epitopes are evidenced with the corresponding color. (**B**) Sandwich ELISA with several pairs of mAbs for the detection of scalar doses of the recombinant spike protein. In the legends, the first indicated mAb is used as coating, and the second indicated mAb is used as secondary antibody. Error bars indicate the standard deviations of technical duplicates from a representative experiment repeated twice.

**Figure 5 biomedicines-11-00610-f005:**
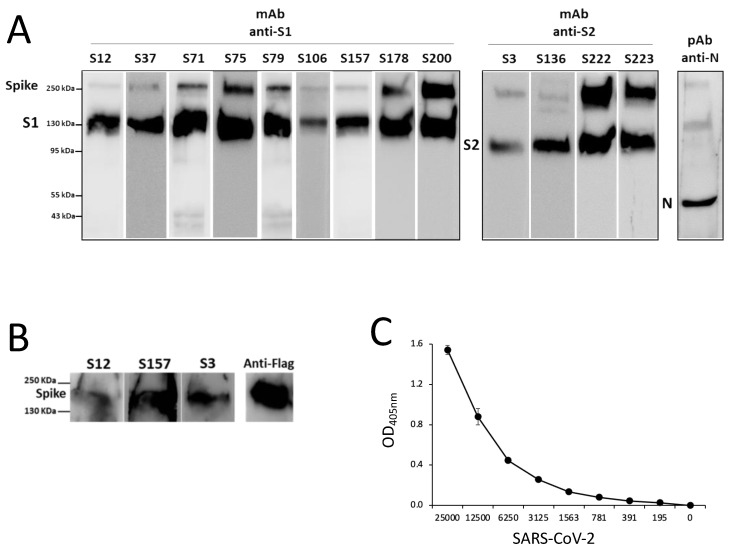
**WB analysis of anti-S1 and anti-S2 mAbs.** (**A**) Supernatants of SARS-CoV-2-infected Vero E6 cells subjected to SDS-PAGE and WB. Polyclonal antibody against SARS-CoV N protein was used as a positive control. One experiment representative of three is shown. (**B**) WB analysis of anti-S1 mAbs (S12 and S157), anti-S2 mAb (S3), and anti-Flag, the latter as a control of transfection; SDS-PAGE of cell lysates from HEK293T-expressing SARS-CoV spike protein followed by WB is shown. One experiment representative of three is shown. (**C**) Sandwich ELISA with S71–S79 pair of mAbs for the detection of scalar quantities of SARS-CoV-2 (PFU/well). Error bars indicate the standard deviations of technical duplicates from a representative experiment repeated twice.

**Figure 6 biomedicines-11-00610-f006:**
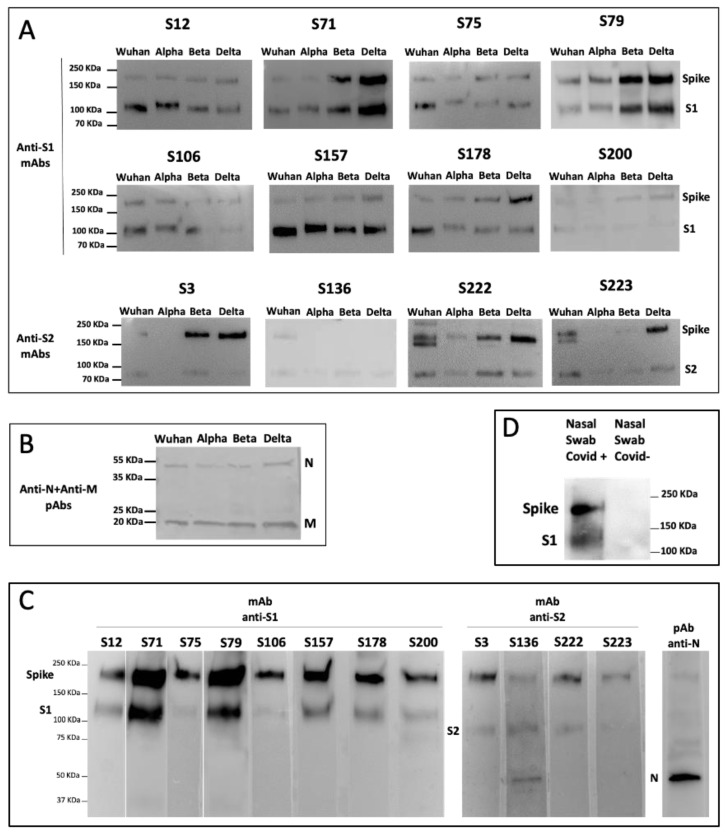
**Isolated S-specific mAbs recognize the major SARS-CoV-2 Variants of Concern.** (**A**) WB analysis of all of the generated mAbs specific for the S1 and S2 domains in SDS-PAGE supernatants of Vero E6 cells infected with SARS-CoV-2 Wuhan, Alpha, Beta, and Delta VOC. The signal was detected by HRP chemiluminescent substrate except for the S200 and S136 mAbs, which were revealed by TMB colorimetric substrate. (**B**) Representative WB of anti-N and anti-M polyclonal Abs (pAbs) detected by colorimetric substrate. (**C**) WB analysis of all of the generated mAbs specific for S1 and S2 proteins in SDS-PAGE supernatants of Vero E6 cells infected with SARS-CoV-2 Omicron VOC. WB of anti-N polyclonal Ab (pAb) was performed as a control on the same strip of S106 mAb. (**D**) WB analysis of a mixture of the two representative mAbs specific for the S1 domain (S71 and S79) in SDS-PAGE nasal swabs from a positive (Covid+) or negative (Covid−) individual. One experiment representative of three is shown.

**Figure 7 biomedicines-11-00610-f007:**
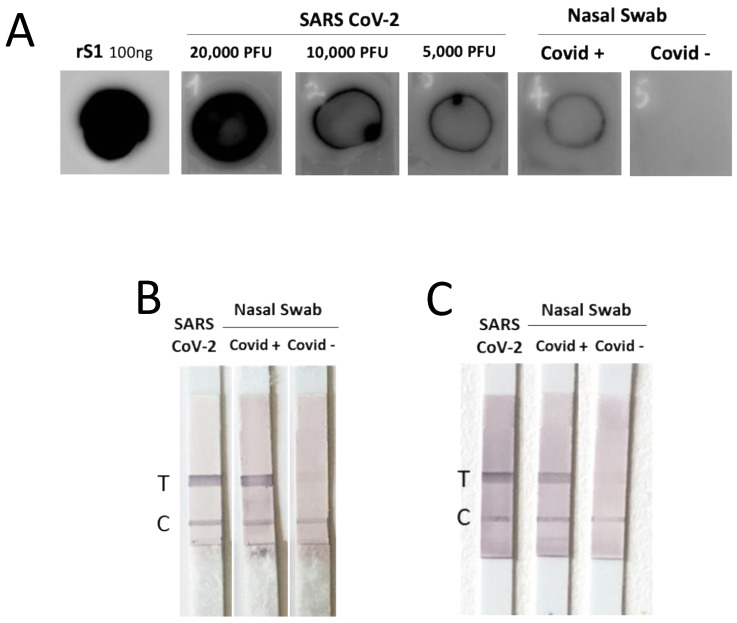
**A preliminary proof-of-concept for the possibility to use the isolated mAbs are tools for an antigenic diagnostic assay.** Dot blot assay of mAb specific for the S1 domain (S71). On PVDF membrane recombinant rS1, supernatant of Vero E6 cells infected with SARS-CoV-2 or nasal swabs from COVID-19-positive patient (Covid+) and from a control healthy subject (Covid−). (**A**) One experiment representative of three is shown. (**B**,**C**) Lateral flow assay with two pairs of anti-S1 mAbs to detect the antigen in the supernatant of Vero E6 cells infected with SARS-CoV-2 or nasal swabs from a COVID-19-positive patient (Covid+) and from a control healthy subject (Covid−). In B, the capture mAb is S71 and the detector mAb is S79; in C, the capture mAb is S71 and the detector mAb is S12.

**Table 1 biomedicines-11-00610-t001:** List of primers used for the generation of recombinant proteins.

	Primers Sequence 5′-3′
**S1 *E. coli* fw**	GCGCGGATCCCAGTGCGTGAACCTGACCACTA
**S1 *E. coli* rv**	CGCGAAGCTTAGCGAGGGGAGTTAGTCTGGGT
**S2 *E. coli* fw**	GCGCGGATCCTCACTGGGTGCTGAGAACTCC
**S2 *E. coli* rv**	GCGCAAGCTTTTAGGACTCGTTCAGGTTCTTGGC
**S1 HEK fw**	GGAAGCTTCGTGAATCTGACAACTCGG
**S1 HEK rv**	GGCTCGAGTCCTTGGAGAGTTTGTCTGGG
**S2 HEK fw**	GGAAGCTTACGGAGCGTGGCATCCCAG
**S2 HEK rv**	GGCTCGAGTCTCCTTCTGGATGTTCACCACGG

**Table 2 biomedicines-11-00610-t002:** List of synthetic peptides of 40 aa in length (with 20 aa overlaps between sequential peptides), covering the S1 domain and the N-terminal portion of the S2 domain in order to finely characterize the epitope recognized by our anti-S mAbs.

Peptide #	Amino Acid Position and Sequence
**1**	1 mfvflvllpl vssqcvnltt rtqlppaytn sftrgvyypd 40
**2**	21 rtqlppaytn sftrgvyypd kvfrssvlhs tqdlflpffs 60
**3**	41 kvfrssvlhs tqdlflpffs nvtwfhaihv sgtngtkrfd 80
**4**	61 nvtwfhaihv sgtngtkrfd npvlpfndgv yfasteksni 100
**5**	81 npvlpfndgv yfasteksni irgwifgttl dsktqslliv 120
**6**	101 irgwifgttl dsktqslliv nnatnvvikv cefqfcndpf 140
**7**	121 nnatnvvikv cefqfcndpf lgvyyhknnk swmesefrvy 160
**8**	141 lgvyyhknnk swmesefrvy ssannctfey vsqpflmdle 180
**9**	161 ssannctfey vsqpflmdle gkqgnfknlr efvfknidgy 200
**10**	181 gkqgnfknlr efvfknidgy fkiyskhtpi nlvrdlpqgf 220
**11**	201 fkiyskhtpi nlvrdlpqgf saleplvdlp iginitrfqt 240
**12**	221 saleplvdlp iginitrfqt llalhrsylt pgdsssgwta 260
**13**	241 llalhrsylt pgdsssgwta gaaayyvgyl qprtfllkyn 280
**14**	261 gaaayyvgyl qprtfllkyn engtitdavd caldplsetk 300
**15**	281 engtitdavd caldplsetk ctlksftvek giyqtsnfrv 320
**16**	541 fnfngltgtg vltesnkkfl pfqqfgrdia dttdavrdpq 580
**17**	561 pfqqfgrdia dttdavrdpq tleilditpc sfggvsvitp 600
**18**	581 tleilditpc sfggvsvitp gtntsnqvav lyqdvnctev 620
**19**	601 gtntsnqvav lyqdvnctev pvaihadqlt ptwrvystgs 640
**20**	621 pvaihadqlt ptwrvystgs nvfqtragcl igaehvnnsy 660
**21**	641 nvfqtragcl igaehvnnsy ecdipigagi casyqtqtns 680
**22**	661 ecdipigagi casyqtqtns prrarsvasq siiaytmslg 700
**23**	681 prrarsvasq siiaytmslg aensvaysnn siaiptnfti 720
**24**	701 aensvaysnn siaiptnfti svtteilpvs mtktsvdctm 740

**Table 3 biomedicines-11-00610-t003:** IgG subclass, spike domain specificity, EC_50_ and peptide recognition of isolated mAbs.

mAb	Subclass	Spike Domain	EC50 (ng/mL)	Epitope Position (AA)
**S200**	IgG1	S1	89.24	21–40
**S37**	IgG1	S1	301.59	281–300
**S75**	IgG1	S1	116.71	301–320
**S106**	IgG1	S1	168.80	301–320
**S71**	IgG2b	S1	86.04	550–580
**S79**	IgG2a	S1	103.81	550–580
**S178**	IgG2b	S1	202.34	550–580
**S12**	IgG1	S1	66.87	661–682
**S157**	IgG1	S1	89.32	661–682
**S222**	IgG1	S2	19.33	686–700
**S223**	IgG1	S2	18.79	686–700
**S136**	IgG1	S2	52.32	686–700
**S3**	IgG1	S2	45.65	721–1182

## Data Availability

The raw data supporting the conclusions of this article will be made available by the authors without undue reservation.

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
