# Peer review of "New Monoclonal Antibodies Specific for Different Epitopes of the Spike Protein of SARS-CoV-2 and Its Major Variants: Additional Tools for a More Specific COVID-19 Diagnosis"

_biomedicines, 2023, doi:10.3390/biomedicines11020610_

Round 1
Reviewer 1 Report
In this manuscript, Mariotti et al., isolated mAbs from mice immunized by purified SARS-CoV-2 S protein. They specified some mAbs which had strong binding affinity with S protein. I have some comments for this manuscript.
Major
1. The journal required that all experimental samples and controls used for one comparative analysis should be run on the same blot/gel image. The authors should check the instruction of journal and repeat the WB.
2. In the introduction some references are missing (line 73, 79, 82, 100). Please add them.
3. Figure 1E bottom: The resolution is poor and it is hard to see the text. In addition, the authors should explain more (detail) what Figure E bottom shows.
4. Figure 1B and Table 3: Some samples has different EC50 and WB intensity. For example, S200 mAb has nice EC50, but WB showed weak bend intensity. Can the authors explain why? It might be better to measure the band intensity and compare it with EC50.
5. Figure 1C: mAbs except S71, S178 and S79 does not reach >2 (OD405nm). Is it an issue with the assay itself? Or does it indicate that the binding affinity are weak?
6. Supplementary Figure 2 left top: the band (SARS-COV-2) is weird and it is hard to distinguish between Full length and S1. Can the authors replace it to new one?
7. Figure 7 and material method: Do the authors know/speculate with which variants the donors were infected? If so, please add the information in the manuscript.
Author Response
Point by point response to reviewers’ comments:
Reviewer 1
In this manuscript, Mariotti et al., isolated mAbs from mice immunized by purified SARS-CoV-2 S protein. They specified some mAbs which had strong binding affinity with S protein. I have some comments for this manuscript.
Major
- The journal required that all experimental samples and controls used for one comparative analysis should be run on the same blot/gel image. The authors should check the instruction of journal and repeat the WB.
Response: We agree with reviewer 1 when she/he reminds that the Journal requires that all experimental samples and controls used for one comparative analysis should be run on the same blot/gel image. Indeed, where a comparative analysis was performed, i.e. binding of a given mAb to different antigens, samples were run in the same gel. In details: Figure 1B shows the correct molecular weight of the soluble forms of recombinant S1, S2 and SΔTM in supernatants of HEK293T cells and Figure 1D shows the right molecular weight of the E. coli recombinant proteins expressed in E coli.: there is no comparative analysis, just a reference to molecular weight markers. Figure 1F shows the consequence of de-glycosylation in terms of MW and the samples incubated for different times with the enzyme were run in the same gel, as well as samples in Fig 3B. The WBs in figures 2B, 5A, 5B and 6C are qualitative analysis to prove the specificity of our mAbs and their ability to recognize the SΔTM (Fig. 2B), SARS-CoV-2 Wuhan isolate (Fig. 5A), SARS-CoV Spike protein (5B) or SARS-CoV-2 Omicron VOC (Fig. 6C): in these figures, the anti-S1 mAb S71 and anti-His are added as a the control of SΔTM expression, the anti-N antibody is the control of the presence of the viruses loaded, and anti-Flag is the control of the transfection of SARS-CoV Spike. The images of all these controls are not shown in comparison to the tested mAbs. On the other hand, Figures 1F, 3B, 6A, 6B and 6D are comparative analysis of the reactivity of each mAb with different antigens and in fact each mAb was tested with antigens run on the same gel as confirmed by the original images that were uploaded as required. No other comparative analyses requiring runs on the same gel were performed. For all these reasons, we are confident to be within the requirements of the Journal as detailed in the “Instruction for authors”.
- In the introduction some references are missing (line 73, 79, 82, 100). Please add them.
Response: We thank the reviewer for noting the missing citations, and the references have now been added.
- Figure 1E bottom: The resolution is poor and it is hard to see the text. In addition, the authors should explain more (detail) what Figure E bottom shows.
Response: We apologize for the poor resolution of the figure 1E bottom. We have now included an image at higher resolution than the previous one and added a detailed description of the Figure 1E bottom content.
- Figure 2B and Table 3: Some samples has different EC50 and WB intensity. For example, S200 mAb has nice EC50, but WB showed weak bend intensity. Can the authors explain why? It might be better to measure the band intensity and compare it with EC50.
Response: We agree with reviewer 1 that some samples have different EC50 and WB intensity. In general, WB analyses were performed to confirm the specificity of the mAbs binding to the right antigens using recombinant antigens or virus lysates, that also contain proteins of cellular origin whose potential reactivity with mAbs cannot be discriminated in ELISA. Moreover, WB were routinely performed in reducing conditions (beta-mercaptoethanol) that may modify the protein conformation and the consequent exposure of epitopes recognized by mAbs, while in ELISA the antigens were used in non-reducing conditions. Therefore, we performed ELISA and WB for different scopes, but not to compare their results in terms of EC50 and WB intensity.
- Figure 1C: mAbs except S71, S178 and S79 does not reach >2 (OD405nm). Is it an issue with the assay itself? Or does it indicate that the binding affinity are weak?
Response: We agree with reviewer 1 that in figure 1C only the mAbs S71, S178 and S79 reach an O.D. >2 (OD405). We usually stopped the enzyme activity when the developing reaction with the substrate reaches a critical color to avoid a plateau effect at high mAb concentrations. Since in the same plate are tested also other mAbs that not necessarily react with the same kinetics, the result is that not all the tested mAbs reaches their highest O.D. value. This may depend on the slightly different mAb concentrations, dilutions or affinity for the antigen. EC50 gives a more reliable measure of mAb affinity than O.D. for our scope, ie. to identify mAbs useful for an efficient antigenic diagnosis of SARS-CoV-2 infection.
- Supplementary Figure 2 left top: the band (SARS-COV-2) is weird and it is hard to distinguish between Full length and S1. Can the authors replace it to new one?
Response: We agree with reviewer 1 that in the supplementary figure the full length S and S1 are nor well separated: we chose that image since the aim of this figure was to show that even with long exposures, no signal of mAbs binding with the other coronaviruses was observable. However, according to the suggestion of the reviewer, we have now modified this figure including an image of the same experiment at lower exposure, that makes easier to distinguish the two bands and uploaded its original image, as requested in the “Instruction for authors” of the Journal. The new figure has now been renamed Supplementary Fig. 3 since a new supplementary figure has been added to reply to the comment of the reviewer 2.
- Figure 7 and material method: Do the authors know/speculate with which variants the donors were infected? If so, please add the information in the manuscript.
Response: In the Material and Methods section, we added that, according to the epidemiology of SARS-CoV-2 variants in the period of samples collection, there was a high probability that the patients could be infected with Omicron variant, even if sequencing was not performed.
Reviewer 2 Report
The current study ‘New Monoclonal Antibodies Specific For Different Epitopes Of The Spike Protein Of SARS-CoV-2 And Its Major Variants: Additional Tools For A More Specific COVID-19 Diagnosis.” determined a rapid need for monoclonal antibodies (mAbs) to detect the SARS-CoV-2 in biological fluids as a rapid tool to identify infected individuals to be treated or quarantined, by targeting S protein since its sequence is more specific for SARS-CoV-2 than N protein. The rS proteins were produced in both the eukaryotic and prokaryotic systems. The set of mAbs that have been generated highlights its unique characteristics and suggests its possible use for antigen detection in rapid or laboratory tests for the diagnosis of infection. The overall study is fundamentally sound. However, before accepting this work, some issues must be resolved by emphasizing the following points.
The introduction requires a more comprehensive with more content to justify the title of the paper. The materials and methods need to be elaborative and explanatory. In Cloning, Expression, and Purification of SARS-CoV-2 proteins in HEK293T cells: the authors should provide a figure to illustrate the entire procedure. In line 325 “For the sandwich ELISA, plates were o/n coated” what does o/n coated indicate?- better write descriptive methods.
The manuscript is missing a detailed conclusion.
Author Response
Point by point response to reviewers’ comments:
Reviewer 2
The current study ‘New Monoclonal Antibodies Specific For Different Epitopes Of The Spike Protein Of SARS-CoV-2 And Its Major Variants: Additional Tools For A More Specific COVID-19 Diagnosis.” determined a rapid need for monoclonal antibodies (mAbs) to detect the SARS-CoV-2 in biological fluids as a rapid tool to identify infected individuals to be treated or quarantined, by targeting S protein since its sequence is more specific for SARS-CoV-2 than N protein. The rS proteins were produced in both the eukaryotic and prokaryotic systems. The set of mAbs that have been generated highlights its unique characteristics and suggests its possible use for antigen detection in rapid or laboratory tests for the diagnosis of infection.
The overall study is fundamentally sound. However, before accepting this work, some issues must be resolved by emphasizing the following points.
- The introduction requires a more comprehensive with more content to justify the title of the paper. The materials and methods need to be elaborative and explanatory.
Response: We have now included in the Introduction section some sentences explaining more accurately the scope of the manuscript. We did not change the materials and methods section, since it is already extremely detailed, but we added a new supplementary figure as required in the following comment.
- In Cloning, Expression, and Purification of SARS-CoV-2 proteins in HEK293T cells: the authors should provide a figure to illustrate the entire procedure.
Response: An explicatory supplementary figure (Supplementary Fig. 1) has now been added to illustrate the entire procedure.
- In line 325 “For the sandwich ELISA, plates were o/n coated” what does o/n coated indicate?- better write descriptive methods.
Response: In line 325 “o/n” has been substituted by “overnight” to indicate the time of incubation of the antigens with the ELISA plates in order to coat the plates wells.
- The manuscript is missing a detailed conclusion.
Response: An additional “5. Conclusion” section has been added.
Round 2
Reviewer 1 Report
The authors answered all my concerns and questions. I think the paper is ready to publish now.